# Central Apnea in Patients with COVID-19 Infection

**Vikram Venkata Puram [1,\*], Anish Sethi [2] ⓘ, Olga Epstein [3], Malik Ghannam [4], Kevin Brown [5], James Ashe [5] and Brent Berry [4,6]**

1   Department of Medicine, Stanford University, Stanford, CA 94305, USA
2   School of Medicine, Drexel University, Philadelphia, PA 19104, USA
3   Department of Internal Medicine, VA Medical Center, Minneapolis, MN 55417, USA
4   Department of Neurology, University of Minnesota, Minneapolis, MN 55455, USA
5   Department of Neurology, VA Medical Center, Minneapolis, MN 55417, USA
6   Department of Neurology, Department of Sleep Medicine, Department of Physiology & Biomedical Engineering, Mayo Clinic, Rochester, MN 55905, USA
\*   Correspondence: vikpuram@stanford.edu

**Abstract:** Background: The Coronavirus Disease 2019 (COVID-19) is a global pandemic that has killed over 1.5 million people worldwide. A constellation of multisystem involvement with SARS-CoV-2 has been reported. COVID-19 has been shown to affect the human nervous system, however, both the extent and severity of involvement have yet to be fully elucidated. In this manuscript, we aimed to better understand the effect of COVID-19 on neuro-respiratory status by studying COVID-19 patients who presented with central apnea. Methodology: We analyzed patient characteristics, clinical outcomes, laboratory results, and imaging results of three patients with symptomatic, PCR-proven COVID-19 and episodes of central apnea. Results: Of the three patients included in this study, two patients developed new central apnea, and one patient developed an exacerbation of underlying central apnea despite COVID-19 treatments with systemic steroids and remdesivir. All occurred, on average, 15 days after the onset of COVID-19 symptoms. At 1-year follow-up, all patients experienced complete resolution of apneic breathing. Conclusions: Physicians should be vigilant for the presentation of COVID-19 with central apnea. Central apnea may be a complication in patients with severe COVID-19 infection. More research is warranted to further understand this association.

**Keywords:** COVID-19; central apnea; neuro-respiratory; case series

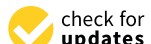

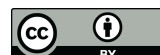

## 1. Introduction

In March 2020, the World Health Organization classified the Coronavirus Disease 2019 (COVID-19) as a global pandemic, which started a few months earlier as a series of unexplained yet highly infectious cases of pneumonia secondary to the novel SARS-CoV-2 coronavirus. The symptoms and downstream effects of COVID-19 spread far and wide beyond the respiratory system [1]. These effects are not only diverse in terms of severity and organ involvement but are yet to be fully discovered. As the world continues to battle this global pandemic, new information is revealed daily as research continues forward in both the clinic and the laboratory.

Though there is still limited evidence that the SARS-CoV-2 virus directly attacks the nervous system, both central and peripheral nervous systems effects have been documented in COVID-19 patients [2]. This is predominantly thought to be due to the secondary effects of being severely ill, suffering from low oxygen levels for prolonged time periods, and multisystem inflammatory reactions. Neurological manifestations such as headaches, anosmia, hypogeusia, ischemic strokes, hemorrhagic strokes, Guillain–Barre Syndrome, neuromuscular problems, and encephalopathy have been documented in over 30% of cases [1,3]. Rarer effects such as seizures, demyelinating disease, vision loss, neuralgia, and ataxia have also been increasingly documented [4–6].

Central apnea has been typically associated with severe illness, particularly processes that affect the breathing centers of the nervous system located in the lower brainstem [7]. This series outlines the first reported cases of central apnea in COVID-19 patients.

## 2. Methods

Patients were selected for this study based on the following inclusion criteria: (1) COVID-19 infection with proven COVID-19 positivity by PCR or serology testing, (2) Clinical diagnosis of central apnea as defined by Gastaut et al. as cessation of breathing for at least ten seconds without other neurologic diagnoses, (3) Admission to the inpatient medicine service for more than 1 day [8].

Three patients were identified that met inclusion criteria (from March 2020 to November 2020): Two patients from the Minneapolis Veterans Affairs Hospital and one patient from Hennepin County Medical Center. All three patients were admitted to their respective hospital's COVID units for comprehensive evaluation and treatment.

The Institutional Review Board (IRB) has been waived for case series at the Hennepin County Medical Center and Minneapolis Veterans Affairs Hospital. Chart review was performed by neurology department physicians at the Minneapolis Veterans Affairs Hospital and the University of Minnesota, Twin Cities. All data were collected retrospectively from electronic medical records, including patient characteristics, physician documentation, imaging studies, and laboratory values. All patients provided consent for participation. All patient health information has been de-identified.

## 3. Results

We report three patients from two large hospital systems with clinically diagnosed central apnea after COVID-19 infection (see Table 1). Case 1 and case 3 represent new-onset central apnea with no previous diagnosis of apnea, while case 2 describes a patient who had self-resolved central apnea from a remote diagnosis over 10 years prior. All patients survived COVID-19 infection and had complete resolution of central apnea at both 3-month and 6-month follow-ups.

**Table 1.** Key patient characteristics, diagnostic evaluations, and hospital course for included patients.

| Variable | Patient 1 | Patient 2 | Patient 3 |
|---|---|---|---|
| Age | Male | Male | Male |
| Sex | 71 | 74 | 47 |
| Central apnea risk factors | Previous stroke | Heart Failure Chronic Obstructive Pulmonary Disease | - |
| Previous apnea diagnosis | No History | Central apnea diagnosed by polysomnography 10 years prior | No History |
| CT Brain Findings | Unremarkable | Unremarkable | - |
| MRI Brain Findings | Chronic infarcts (seen on prior scans) but unremarkable otherwise | - | Unremarkable |
| Cerebrospinal Fluid (CSF) Analysis | - | Normal CSF Panel. CSF not tested for COVID-19 | Elevated Protein to 98. PCR was positive for COVID-19. |
| COVID-19 Symptoms | Cough Shortness of Breath Fatigue Myalgia | Cough Shortness of Breath | Fatigue Myalgia Headache Blurred Vision |
| Time from symptom onset to apneic episode | 14 Days | 12 Days | 20 Days |
| COVID-19 Treatment | Steroids + Remdesivir | Steroids + Remdesivir | Steroids + Remdesivir |

### 3.1. Case 1

A 71-year-old male presented to the emergency department (ER) complaining of one week of generalized weakness, myalgias, fatigue, and lightheadedness with two days of nonproductive cough and mild shortness of breath. The patient had a history of type II diabetes, hypertension, chronic kidney disease stage 3a, peripheral vascular disease, and carotid artery disease with a remote history of prior strokes. Upon evaluation, the patient denied fevers, chills, nausea, vomiting, abdominal pain, and headaches. His oxygen saturation was 93% on room air with normal blood pressure and respiratory rate. Chest X-ray revealed airspace disease in the left lower lobe consistent with a viral infection, and initial labs showed leukopenia and lymphopenia. His nasopharyngeal PCR was positive for COVID. After a few hours, the patient developed hypoxemia with oxygen saturation down to 89% and was admitted to the hospital with a diagnosis of COVID-19 pneumonia. He was treated with remdesivir and dexamethasone, with gradual improvement in his respiratory symptoms. After six days, his symptoms resolved and he was discharged home with instructions to continue dexamethasone for three days and enoxaparin for five days.

Four days after his discharge, the patient returned to the emergency department by ambulance with altered mental status, including loss of alertness, disorientation, and mumbled speech. Narcan 0.4 mg was administered by EMS with no immediate improvement. Upon arrival at the hospital, he underwent extensive evaluation for the etiology of his mental status change. His head CT was negative for acute bleeding and ischemia. He had a negative rapid drug and alcohol screen, baseline basic metabolic panel (BMP) and complete blood count (CBC), improving chest X-ray, and normal urinalysis and procalcitonin. After the initial unrevealing workup, the patient was readmitted to the hospital for further evaluation of delirium. The patient reported complete resolution of his cough and shortness of breath and had an oxygen saturation of 95% on room air. Two days after admission, he was found to be more obtunded with a severely depressed respiratory rate (low of 4 breaths per minute). He was intubated for airway protection and mild hypoxia and was transferred to the ICU. He was extubated the following day and rapidly improved after starting high-dose dexamethasone (five-day burst of 6 mg daily). He was transferred to the COVID-19 rehab unit, where he did very well and was able to discharge home after a total of 19 days in the hospital.

When further investigating the patient's mental status and respiratory depression, brain MRI was ordered and showed chronic infarcts that were unchanged from prior scans but did not show any acute intracranial disease process. Lab tests relevant to causes of encephalopathy were ordered and yielded no abnormalities in sodium, potassium, magnesium, phosphate, mild hyperglycemic in the high 200 s (but not likely enough to cause hyperosmolar hyperglycemic syndrome), normal serum osmolality, liver function tests, and blood urea nitrogen within normal limits making hepatic and uremic encephalopathy unlikely, no thiamine deficiency, no thyroid abnormalities with normal TSH, and no B12 deficiency. CT, MRI, CSF analysis, laboratory value, medication review, nor physical exam findings were able to help explain the etiology of severe respiratory depression, this, it was thought that a post-viral prolonged inflammatory response after patient's recent COVID-19 infection most likely caused an extended encephalopathy resulting in central apnea. At 1-year follow-up, the patient had no further apneic episodes or concern for central apnea.

### 3.2. Case 2

Patient is a 74-year-old with a history of coronary artery disease, heart failure with reduced ejection fraction, ventricular fibrillation, and ventricular tachycardia with ICD implantation 1 year prior, atrial fibrillation on apixaban, hypertension, hyperlipidemia, chronic obstructive pulmonary disease, distant history of alcohol, and cocaine abuse along with known central sleep apnea diagnosed 10 years ago. He presented to urgent care complaining of a few days of shortness of breath, wheezing, and cough without fevers, chills, nausea, vomiting, or chest pains. Patient had normal oxygen saturation on room air but was tachypneic (respiratory rate of 28) and found it difficult to speak more than a few

words. Patient was found to be positive for COVID-19 and was transferred to the hospital for stabilization.

Upon admission, the patient received a chest X-ray which was unremarkable for pneumonia. CT pulmonary angiogram was also ordered and showed no evidence of pulmonary emboli. However, there was central and lower lobe bronchial wall thickening with areas of mucous plugging, and rounded opacities at the left lung base, thought to represent possible atelectasis, inflammation, or infection. Labs were unremarkable other than mild lymphopenia and elevated troponin (thought to be due to demand ischemia in the setting of viral infection). Electrocardiogram was unremarkable for acute coronary syndrome. Patient completed a course of remdesivir and steroids. Patient's breathing slowly improved over these 5 days, and he also had a reduction in the wheezing he experienced on admission after the continuation of his budesonide/formoterol and tiotropium inhalers. Over this time period, the patient's other medical issues were managed appropriately without complications. His lymphopenia and elevated troponin resolved.

On hospital day 10, the patient had an episode of interval breathing where he became apneic for ~30 s and recovered with brief tachypnea. His oxygen saturation was maintained above 90% on room air. Upper and lower extremity pulses were felt bilaterally, and the patient had a blood pressure of 132/72, heart rate of 80, and blood glucose of 129. The patient had his eyes closed and was responsive to only loud verbal cues. This episode lasted 10–15 min, after which he recovered to baseline. With unremarkable EKG, ICD interrogation, and head CT, neurology was consulted. Evaluation for possible seizures, including EEG, was unremarkable. The next day, pulmonology was consulted and determined that the patient's apneic episodes were most likely due to the recurrence of his known central sleep apnea diagnosis 10 years ago, as diagnosed via polysomnography (PSG) testing. After his initial diagnosis, the patient was managed with BIPAP ASV. It was discontinued after 4 years due to its contraindication in systolic heart failure and resolution of his apneic episodes. After discussion with sleep medicine, it was thought that the post-viral state after the COVID-19 infection ultimately caused exacerbation of the patient's dormant underlying central apnea. With the current lack of evidence linking COVID-19 and central apnea, this was primarily a diagnosis of exclusion due to the absence of any other exacerbating factor as well as the suspicious timing of central apnea exacerbation and COVID-19 infection (the patient had been asymptomatic for over 10 years prior to COVID-19 infection). At 1-year follow-up, the patient had no further apneic episodes or concern for central apnea.

*3.3. Case 3*

A 47-year-old male with no medical history presented with 3 weeks of fatigue, myalgia, headache, and blurred vision. The patient was initially lethargic but otherwise had a normal neurological examination. Several days prior to this evaluation, the patient had momentary episodes of hypoxia and shallow breathing without clear precipitant. Evaluation with chest X-ray revealed no infiltrates or obvious abnormalities. He was brought into the hospital for admission and further evaluation. As part of current admission screening protocols, he was tested for COVID-19, which returned positive. Ferritin was elevated to 210, and CRP was elevated to 123. ESR was within normal limits. CSF profile and culture were normal aside from an elevated protein at 98. Cerebrospinal fluid polymerase chain reaction for COVID-19 was positive. MRI Brain with and without contrast was performed and was unremarkable. The patient was initiated on remdesivir and dexamethasone per recommendations from infectious disease.

On the first night of admission, the patient experienced at least six apneic episodes, all of which set off code blue calls. There was suspicion of seizure during these episodes, so electroencephalography monitoring was initiated. During these apneic periods, no seizure or interictal epileptiform discharges were revealed. Neurology recommended spinal axis imaging with contrast which was entirely unremarkable. These apneic episodes continued into the day, and although the patient had no associated altered mental status, he had difficulty with voluntary respirations. With COVID-19 being the only significant change

in the patient's health over the previous few years, neurology administered IV steroids (methylprednisolone 500 mg daily for 5 days) to treat a presumed post-viral inflammatory state. This diagnosis, though primarily a diagnosis of exclusion given the current lack of evidence of COVID-19-induced central apnea, was supported by the patient's elevated inflammatory markers, such as CRP, in the context of unremarkable MRI findings and negative seizure workup. After 1 week, repeat brain MRI and lumbar puncture were performed, and this time, CSF COVID-19 testing was negative. CRP and ferritin returned to normal limits. The patient was referred for PSG testing upon discharge, which revealed no obstructive or central apneas (this was undertaken 3 months after discharge). At 1-year follow-up, the patient had no further apneic episodes or concern for central apnea.

## 4. Discussion

Central apneas are periods of absent airflow due to a markedly reduced or temporary lack of respiratory effort. This occurs when inhibitory input to the respiratory centers of the brain exceeds excitatory input, which is far more likely to occur at night due to lower baseline excitatory activity (hence often known as central sleep apnea) [9,10]. Key to the etiology of central apnea is the interruption to the normal function of pontomedullary pacemaker cells, medullary neurons responsible for respiratory rhythmogenesis [11]. Causes may be genetic (such as a congenital PHOX2B mutation that results in disordered homeostatic central respiratory chemosensitivity) or acquired (such as brainstem lesions, spinal cord disorders, viral encephalitis, or chronic opioid use) [12]. Symptoms including excessive daytime sleepiness, insomnia, inattention, and poor concentration are common. However, insidious onset frequently causes central apneas to go unrecognized. These individuals are prime candidates for in-laboratory PSG, the test that helps confirm a central apnea diagnosis [13]. However, central apnea does not always necessitate PSG for a diagnosis, as decreased respiratory drive in the setting of otherwise normal pulmonary mechanics may be diagnosed clinically, especially when transient in nature. After diagnosis, patients are typically treated with positive airway ventilation to breathe, most often with Bilateral Positive Airway Pressure (BiPAP) [14]. Adaptive servo-ventilation (ASV) with auto-adjusting pressure support and drugs, such as acetazolamide, are sometimes used as well [15,16].

Currently, COVID-19 does not have a clearly elucidated mechanism for how it affects pontomedullary pacemakers and subsequently disrupts the activation of inspiratory thoracic muscles. However, we propose two separate hypotheses on this mechanism. First, we propose a direct mechanism in which the COVID-19 virus (SARS-CoV-2) directly causes dysfunction of pontomedullary pacemakers. SARS-CoV-2 is a relatively new virus with much of the scientific research still yet to be conducted. However, it has a high level of homology to SARS-CoV (cause of the 2003 SARS outbreak), which has been clearly shown to affect medullary neurons in both animal studies and human autopsies. A study of mice transgenic for human ACE2 (the SARS-CoV receptor) by Netland et al. showed that severe dysfunction and/or death of infected neurons, especially those located in cardiorespiratory centers in the medulla, was a cause of animal death [17]. Coronaviruses, particularly SARS-CoV, have been shown to cause direct infection of the human CNS, consistent with detection of SARS-CoV in the neurons of human autopsies [17,18]. Early evidence has also supported the neuroinvasive potential of COVID-19 [19]. The nervous system is a new territory of COVID-19 research, and even with the high similarity between SARS-CoV and SARS-CoV-2, research has yet to fully confirm whether the potential invasion of SARS-CoV-2 may be partially responsible for CNS-mediated acute respiratory failure in COVID-19 patients.

Secondly, we propose an indirect mechanism in which the COVID-19 inflammatory state causes dysfunction of brainstem neurons, which contributes to respiratory distress, either with or without concurrent neuronal invasion. Dysregulation of cytokines/chemokines and the resulting "cytokine storm" of excessive inflammatory responses to COVID-19 infection have been well-established as key factors behind worsened patient outcomes [20]. The three patients discussed above represent cases of COVID-19-induced adult central apnea. In

cases 1 and 3, COVID-19 infection caused new-onset central apnea in a patient with no history of it, while in case 2, infection caused a recurrence of a remote history of central apnea. COVID-19 has been characterized by three stages: (1) Early infection, (2) pulmonary phase, (3) inflammatory phase [21]. The inflammatory phase is a post-viral phase of systemic hyperinflammation shown to affect multiple organs, which occurs 7–15 days from onset of symptoms, likely involving both direct viral invasion as well as pro-inflammatory "cytokine storm" [20]. This phase serves as the foundation for the idea that anti-inflammatory drugs, such as corticosteroids, may have a beneficial effect. Consistent with this, the average time from symptom onset to first apneic attack for these three patients is 15 days, which falls at the very end of the stage 3 hyperinflammation period (See Figure 1). The success of using steroids to reduce COVID-19-driven hyperinflammation has been validated but is still anecdotal in many cases. Awareness of this link between the COVID-19 inflammatory response and centrally driven respiratory distress may have a strong guiding significance for the prevention and treatment of COVID-19-induced respiratory failure. We believe that the threshold for treatment with steroids, as in these three patient cases, should be low.

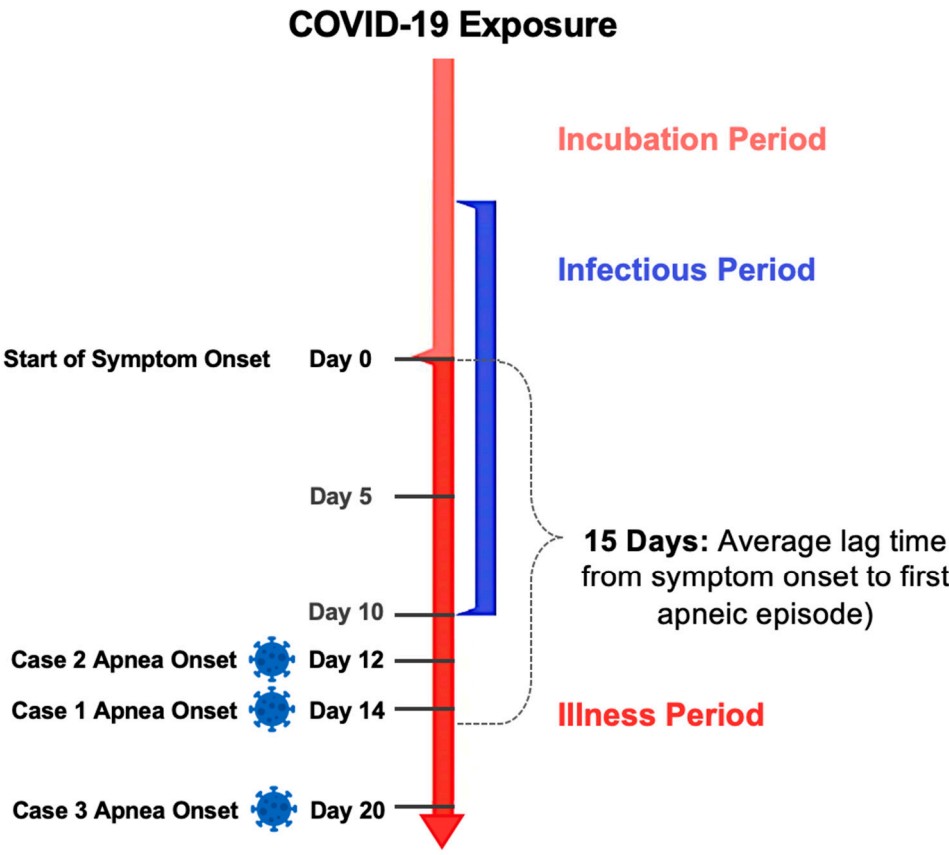

**Figure 1.** Timeline of events after COVID-19 exposure. Onset of symptoms marked as day 0. Length of infectious period for the average COVID-19 patient is 10 days after onset of symptoms, as per current CDC guidelines. Time from onset of symptoms to first apneic episode was 14 days, 12 days, and 20 days for case 1, case 2, and case 3, respectively.

Central apnea caused by COVID-19 has never been described in the literature in adult patients, however, there has been a report of central apnea in a pediatric patient with COVID-19. Enner et al. outline the case of a healthy 14-year-old girl with no prior medical history that developed central apnea ~1 week after the onset of respiratory symptoms and fever [22]. Central apnea is a relatively rare disease with an incidence rate of ~0.9%. Its relationship with COVID-19 remains largely unknown, both with respect to new-onset central apnea as well as exacerbations of preexisting central apnea [23–25]. Knowledge of this association is of high importance, given the possible morbidity and mortality associated

with central apnea, especially in patients with concomitant heart failure. The novelty of our paper is in elaborating for the first time the association between COVID-19 and central apnea in adult patients. This paper primarily aims to raise awareness of this newly discovered relationship and also to warn neurologists, pulmonologists, hospitalists, and other healthcare providers caring for COVID-19 patients of this potentially dangerous and even life-threatening complication that may arise from COVID-19 infection. However, our study has some limitations. First, the evidence is based on a small series of three patients, which limits its generalizability. Secondly, although central apneas may be diagnosed clinically, as outlined by Gastaut et al. in 1966, a well-established diagnostic criteria used widely prior to the invention of PSG, more recent clinical definitions have employed PSG testing in addition to observations of breathing cessation to offer a stronger diagnosis. [8] Immediate PSG was not possible in these cases due to the acuity of patient illness, hospital limitations for PSG testing to be completed on an outpatient basis and hospital infectious precautions related to COVID-19. Additionally, symptoms had resolved by the time patients were sufficiently recovered to undergo PSG. Future studies involving larger patient cohorts and formalized PSG testing are recommended to further examine this association between COVID-19 and central apnea.

## 5. Conclusions

Infection of humans with COVID-19 has resulted in significant morbidity and mortality, with respiratory failure being the primary cause of death. The neurological effects of COVID-19, though still unclear at this time, have been shown in numerous patient cases. This case series outlines the clinical presentation, workup, diagnosis, and treatment of three patients with episodes of clinically diagnosed central apnea after COVID-19 infection. Two patients experienced new-onset central apnea, while one patient experienced a recurrence of previously treated central apnea. The average time of COVID-19 symptom onset to the start of apneic episodes was 15 days. None of these patients had a recurrence of apneic episodes after COVID-19 treatment. Central apnea has not been previously described in adult patients with COVID-19, though previous coronaviruses such as SARS-CoV have well-established evidence linking infection to respiratory depression both in animal studies and human autopsies. The safety and efficacy of steroid use in the COVID-19 inflammatory state have already been well-established, and thus, we recommend a low threshold for starting steroids in COVID-19 patients with suspected apneic episodes to prevent the dangerous consequences and even death that may occur from COVID-19-induced central apnea.

**Author Contributions:** B.B., K.B., O.E., V.V.P. responsible for the clinical management of the patient. V.V.P., A.S., M.G., J.A. responsible for drafting the manuscript. All authors have read and agreed to the published version of the manuscript.

**Funding:** This research received no external funding.

**Informed Consent Statement:** Informed consent was obtained from all subjects involved in the study. No identifying patient information was used.

**Data Availability Statement:** All the data supporting our findings is contained within the manuscript.

**Acknowledgments:** The authors thank the patients who generously agreed to participate in this medical report.

**Conflicts of Interest:** The authors declare that they have no competing interest.

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
