# Peer review of "Central Apnea in Patients with COVID-19 Infection"

_2571-8800, doi:10.3390/j6010012_

Round 1

Reviewer 1 Report

Methods

- A detailed description of the status of case 1 and 2 is made

- Case 3 shows the completion of the PSG three months after discharge. Cases 1 and 2 were not reviewed later by PSG? Because?

Discussion

- The hypotheses of the mechanisms are well justified

- It could go a little deeper into the three stages, based on the evidence of clinical results obtained

- Describe the novelty of the work by associating COVID-19 with central apnea

Conclusions

- Although the conclusion is interesting, it is necessary to delve deeper into the dangerous consequences that can occur due to central distress induced by COVID-19

- Future challenges should also be included in this relationship of COVID-19 vs. Central Apnea, with possible investigations to be carried out, larger cohorts and the measurement of PSG.

- Within the inclusion criteria, a review of previous history of apnea in patients and previous neurological events could be included.

References

- They are current and in large quantity, which generates a good theoretical support

Objective

- The objective set out in the paper is "To better understand the effect of COVID-19 on the neurorespiratory state...", which is not fully met given the small cohort of patients, however, a detailed description of the effects stands out and that this topic should be further studied.

General

- The manuscript opens a new research question, associated with the covid and the presence of Central Apnea, which is why it is a paper of considerable interest.

- The paper has a solid support in the background

Reviewer 2 Report

Central Apnea in patients with COVID-19 Infection

Feedback using CARE Checklist (https://www.care-statement.org/):

Item 1: "Case report" required in article title

Item 2: "Case report" required in Keywords

Item 3b: Abstract: Missing symptom/diagnosis information (e.g. COVID symptoms/diagnosis)

Item 3c: Abstract: Missing therapeutic interventions and outcomes

Item 9: Exact types and dosing of therapeutic interventions appear lacking for patients (e.g. Line 182 "neurology recommended five days of IV steroids".  Where steroids given? What drugs exactly and what dosage?)

Item 10: Was there any patient follow up?

Item 11: Strengths AND limitations of this case report needed in discussion.

Other comments:

Line 33: "novel SARS-CoV-2 coronavirus."

Citation needed

Line 42: "Neurological manifestations such as headaches, anosmia, hypogeusia, ischemic strokes, hemorrhagic strokes, Guillain-Barre Syndrome, neuromuscular problems, and encephalopathy have been documented in over 30% of cases [3]"

More citations needed.
